# Transparency- and Repellency-Enhanced Acrylic-Based Binder for Stimuli-Responsive Road Paint Safety Improvement Technology

**DOI:** 10.3390/ma14226829

**Published:** 2021-11-12

**Authors:** Won-Bin Lim, Ji-Hong Bae, Gyu-Hyeok Lee, Ju-Hong Lee, Jin-Gyu Min, PilHo Huh

**Affiliations:** Department of Polymer Science and Engineering, Pusan National University, Busan 46241, Korea; wblim@pusan.ac.kr (W.-B.L.); jihong.bae@pusan.ac.kr (J.-H.B.); ghl8314@pusan.ac.kr (G.-H.L.); dlwnghd15@pusan.ac.kr (J.-H.L.); jgmin@pusan.ac.kr (J.-G.M.)

**Keywords:** acrylic polymer, stimuli responsive polymer, external stimuli-responsive material, photostimulation, binder

## Abstract

In the current study, an acrylic polymer binder applicable to road signs was successfully developed by mixing various acrylic, acrylate-type, and photoinitiator-based monomer species at different acrylate series/silicone acrylate ratios. An amorphous acrylic monomer was used, and the distance between the polymers was increased to improve transparency. The binder was designed with the purpose of reducing the yellowing phenomenon due to resonance by excluding the aromatic ring structure, which is the main cause of yellowing. The optical properties of the binder were determined according to the content of n-butyl methacrylate/methyl methacrylate and the composition of the crosslinking agent in the formulation. Allyl glycidyl ether and dilauroyl peroxide were used to improve the yellowing problem of benzoyl peroxide, an aromatic photoinitiator. Adding a silicone-based trivalent acrylic monomer, 3-(trimethoxysilyl)propyl methacrylate (TMSPMA), was also found to have a significant effect on the transparency, shear properties, and water resistance of the binder. When 15 wt% TMSPMA was added, the best water repellency and mechanical properties were exhibited. The surface morphology of the improved binder and the peeling part were confirmed using field emission scanning electron microscopy. The acrylic polymer developed in this study can be applied in the coating and adhesive industries.

## 1. Introduction

External stimuli-responsive material has an intelligent function in that it recognizes physicochemical stimuli and responds by itself. These technologies are being used in the information technology (IT), automotive, packaging, and road facility industries as well as end-user industries. A continuously increasing amount of research and development of smart polymers and stimuli-responsive polymers, such as the manufacturing of biomedical application materials using heat sensitive materials [1,2,3,4,5], external stimuli-responsive materials and application technology [6,7,8], smart polymer materials with nanostructured surfaces for biomolecular recognition and bonding [9,10,11], and smart polymer materials sensitive to temperature and light, is being carried out. As interest in smart materials is rapidly increasing in the electronics and automobile industries, automobiles, smart displays, energy harvesting, sensor devices, and smart coating materials are also representative research fields [12,13]. Road marking technology should have the characteristics of phosphorescence, fluorescence, and retroreflection by photostimulation (sunlight, streetlamp lighting, automobile headlight, etc.) [14,15,16,17]. Luminescent paints for improving visibility at night can be applied to areas without lighting and vulnerable sections of roads. They can be used in special-purpose road marking, such as in high-risk accident area marking and parking lot line marking and can be of great assistance in preventing traffic accidents at night.

This study develops a high-transparency acrylic polymer binder base material to protect road paint. The transparency of the base resin binder is a prerequisite to maximize the emission of phosphorescent and fluorescent materials. Binder molecular formulation for adhesion and water resistance was designed with consideration of the environmental and mechanical factors of the road surface. The acrylic polymer binder prepared in this formulation was analyzed using the universal testing machine (UTM, LS Series, AMETEK, Inc., Berwyn, IL, USA), contact angle meter (SEO, Phoenix300, Suwon, Korea), ultraviolet(UV)–Vis spectrophotometer (Scinco, Mega-800, Seoul, Korea), and field emission scanning electron microscopy (FE-SEM, SUPRA 25, Carl Zeiss AG, Oberkochen, Germany).

## 2. Materials and Methods

### 2.1. Materials of Stimuli-Responsive Acrylic Polymer Binder

The acrylic polymer binder was synthesized using polyethylene glycol diacrylate (PEG400DA, Mn = 508 g/mol, Miwon Chemicals Co., Ltd., Ulsan, Korea), 2-hydroxyethyl methacrylate (2-HEMA, Merck KGaA, Darmstadt, Germany), and methyl methacrylate (MMA, Merck KGaA, Darmstadt, Germany) as its main components. The contents of ally glycidyl ether (AGE, Merck KGaA, Darmstadt, Germany) and n-butyl methacrylate (BMA, Merck KGaA, Darmstadt, Germany) as acrylic monomer species were controlled. 3-(trimethoxysilyl)propyl methacrylate (TMSPMA, Merck KGaA, Darmstadt, Germany) as a trivalent silicone acrylic monomer, benzoyl peroxide (BPO, Jeongseok Chemical Co., Ltd., Daejeon, Korea), and dilauroyl peroxide (DP, Merck KGaA, Darmstadt, Germany) as curing agent with nitrogen catalyst (PTE, Jeongseok Chemical Co., Ltd., Wanju, Korea) were mixed, changing the ratio of acrylates and silicone acrylates [18,19,20]. The mixed materials were evaluated by making samples via curing at room temperature [21,22,23]. Figure 1 presents a schematic diagram of procedure used to form acrylic polymer binder. Table 1 presents the split composition of the comparative groups in molar ratio.

### 2.2. Characterization

The shear strength was measured using specimens of the following specifications: Based on the ASTM D1002 method, the specimens used to measure the shear strength were manufactured according to the dimensions of 2.5 cm × 10.2 cm × 0.3 cm by using an applicator on a steel plate. The prepared samples were stored at room temperature for 24 h before use. Specimens were used to evaluate the adhesive properties of the binder, and universal testing machine was used. Evaluations were performed at room temperature and tested at a loading speed of 1.3 mm/min. Each sample was repeated 5 times, and the mean was determined. Samples for measuring transmittance and contact angle were measured by applying an applicator to a glass substrate with a thickness of 175 μm. Transmittance was measured in a wavelength range of 300 nm to 700 nm at room temperature using a UV spectrophotometer. To measure the gel rate, 5 g of binder was uniformly dispersed using a spin coater at 2000 rpm to make a sample. Gel rate of the binder was confirmed using the weight measurement method. Then, 5 g of the uniformly polymer film was weighed and placed in a 20 mL vial bottle, and 15 g tetrahydrofuran (THF, Merck KGaA, Darmstadt, Germany) was injected as a solvent, shaken for 1 min, left for 24 h, and then shaken for 1 min. This process was repeated 3 times. Thereafter, it was filtered through a vacuum pump, and the sample was dried at room temperature for 2 h and then oven dried at 70 °C. Contact angle of the binder was measured using a contact angle meter to measure the static contact angle with water at room temperature.

## 3. Results and Discussion

In order to improve yellowing and transmittance, BPO and BMA were adjusted to form a compounding ratio. First, Figure 1a,b show that the transmittance was improved by adding AGE, except for BMA content, which is widely used in binders. To improve the yellowing phenomenon prominent in Figure 1b, in the second experiment, the curing agent was changed from BPO to DP (Figure 1c–f). To increase the mechanical properties after changing the curing agent, the mixing ratio was established by adding trivalent acrylate to the existing acrylate monomer composition. As the sample in Figure 1d–f shows, it can be confirmed that transparency and yellowing improved by adding TMSPMA.

As a result of adding a silicone-containing acrylate material (5~25 wt%) to the acrylic binder, it was confirmed that the formulation containing 15 wt% TMSPMA had the best transparency and shear properties, highest gel rate, and excellent water resistance. When the content of 15 wt% TMSPMA was added, the stress improved due to the inorganic and chemical bonding caused by the addition of the silane coupling agent. Si-O cross-linking improves adhesion, but excessive TMSPMA content reduces the maximum shear force due to flexible Si-O bonding. When the TMSPMA content was 15 wt%, the highest gel rate was observed. With a high gel fraction, more amorphous structures formed, which contributed to improved transmittance. Finally, through the contact angle measurement using water, it was confirmed that when 15 wt% TMSPMA was added, the best water repellency could be observed. Figure 2 shows a graph displaying the changes that occurred with the addition of TMSPMA, and detailed numerical values are indicated in Table 2.

As shown in Figure 3, the difference in shear strength according to the content of TMSPMA was confirmed by observing the cross-sectional morphology. The interaction with the bonding surface is influenced by factors such as material compatibility, solubility parameters, an appropriate material ratio, and entrapment of moisture or gas during the reaction. When the binder was made by adding 15 wt% TMSPMA to the mixture, the highest gel rate was achieved. So, 15 wt% TMSPMA is considered to be the optimal amount to achieve the best water repellence when mixed with other acrylates. When 15 wt% TMSPMA was added, wrinkles of a thicker stem formed, and it is assumed that water repellence increased as a result of the high crosslinking density. It was confirmed through the cross-sectional morphological structure that the Si-O cross-link formed due to the addition of silicone-based acrylate contributing to the increase in adhesion.

The added TMSPMA silane material has a structure of X-Si-(OCH_3_)_3_. Since the bond of -(OCH_3_) is relatively more stable than the bond dependent on the X functional group, the reaction with organic resin occurs at the X functional group. The alkoxy group (-OR) forms a chemical bond with inorganic materials (glass and metals) to improve the adhesion strength. Si-O cross-linking affects the adhesive strength by increasing the gel fraction. Figure 4 shows binder interfaces after detachment to the adherend. The binder with the silicone-based acrylic monomer led to the effective interface bonding and good tensile properties. In the surface peel-off test, the breakage of the interface was observed to be significantly reduced.

## 4. Conclusions

An acrylic polymer binder with high transmittance and adhesive strength was developed by using a silicone-based trivalent acrylic monomer, 15 wt% TMSPMA. Since the adhesive exists in liquid form, it has fluidity when it comes into contact with the adherend. In this way, the adhesive gradually hardens while in contact with the adherend and has adhesive strength. It changes to a solid via heat or chemical reaction and resists peeling by bonding with the interface. In order to improve the adhesion, it is effective to make the chemical bond between the adherend and the adhesive stronger. As the content of the trivalent or higher polyfunctional acrylate increases, its curing time and adhesive strength increase as a result of crosslinking. To obtain high transparency, an amorphous structure is used, so the chain distance between the polymers is wide. The yellowing phenomenon is solved via the removal of the aromatic ring and optimization of the initiator. It is confirmed in this experiment that the binder made via the addition of TMSPMA, a trivalent silicone acrylate, is best formulated at 15 wt%. Highly transparent binder technology is expected to be applied to the fields of transportation (in intelligent base resins), visibility of night roads, and pedestrian safety.

## Data Availability

The data presented in this study are available on request from the corresponding author.

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
