# Peer review of "Transparency- and Repellency-Enhanced Acrylic-Based Binder for Stimuli-Responsive Road Paint Safety Improvement Technology"

_materials, 2021, doi:10.3390/ma14226829_

Round 1
Reviewer 1 Report
The article titled “Transparency-enhanced external stimuli-responsive material for road traffic safety improvement technology” shows the modification of synthesis that results in obtain more transparent and colourless material. The subject is important from application point of view. Articles need major revision. My comments and suggestions are listed below.
In my opinion the introduction is unsatisfactory for example does not present up-to-date materials that are used in this type of application, the cited literature should be extended.
The Authors called materials stimuli responsive, even in title. What kind of stimuli responsiveness exhibits this material? How was it tested?
The scheme of adhesive test in Figure 2 (A) should be corrected, it is barely visible now.
Why is water repellence the best for addition 15 wt.% of TMSPMA? The structure of the materials imaged on Figure 3 should be describe in more detail.
In sentence: “The added TMSPMA silane material has a structure of X-Si(OH)2-OR, so the X functional group reacts with the organic resin. “ Rather structure should be shown as X-Si-(OR)3, or after hydrolysis X-Si(OH)3, or if not how is it known that only two groups hydrolyse?
I don’t find this sentence clear: (from Conclusions) “The adhesive hardens in the form of a liquid that can be easily wetted by contact with the adherend when attached, and it has adhesive strength.”
Author Response
I have attached the file with an answer to the reviewer's question, so please check it.

Reviewer 2 Report
The manuscript titled “Transparency-enhanced external stimuli-responsive material for road traffic safety improvement technology” analyzed the influence of the addition of 3-((trimethoxysilyl)propyl methacrylate (TMSPMA) in an acrylic polymer binder. The authors studied mechanical properties, the transparency of the binder, and the water repellency. The results are promising, well explained, and justified, so I recommend the publication of this manuscript in materials. Before its publication, the manuscript should be revised in some points, mainly in describing the experiments performed. The authors did not explain how they cured the samples for each experiment. We only know that they prepared a film by spin-coated for shear strength measurements. They did not mention the cure temperature or thickness of the films and whether they were also used for transmittance, gel rate calculation, and contact angle. After this improvement of the experimental section, this manuscript is suitable for materials.
Author Response

(The authors gave the same response as above.)

Round 2
Reviewer 1 Report
Dear Authors,
I found replies to be enough, and I accept article for publication.